# Simulation of Traffic-Born Pollutant Dispersion and Personal Exposure Using High-Resolution Computational Fluid Dynamics

Sadjad Tajdaran [1,*], Fabrizio Bonatesta [1], Byron Mason [2] and Denise Morrey [1]

[1] School of Engineering, Computing and Mathematics, Oxford Brookes University, Wheatley Campus, Oxford OX33 1HX, UK; fbonatesta@brookes.ac.uk (F.B.); dmorrey@brookes.ac.uk (D.M.)

[2] Department of Aeronautical and Automotive Engineering, Loughborough University, Loughborough LE11 3TU, UK; b.mason2@lboro.ac.uk

[*] Correspondence: stajdaran@brookes.ac.uk

**Abstract:** Road vehicles are a large contributor to nitrogen oxides (NOx) pollution. The routine roadside monitoring stations, however, may underrepresent the severity of personal exposure in urban areas because long-term average readings cannot capture the effects of momentary, high peaks of air pollution. While numerical modelling tools historically have been used to propose an improved distribution of monitoring stations, ultra-high resolution Computational Fluid Dynamics models can further assist the relevant stakeholders in understanding the important details of pollutant dispersion and exposure at a local level. This study deploys a 10-cm-resolution CFD model to evaluate actual high peaks of personal exposure to NOx from traffic by tracking the gases emitted from the tailpipe of moving vehicles being dispersed towards the roadside. The investigation shows that a set of four Euro 5-rated diesel vehicles travelling at a constant speed may generate momentary roadside concentrations of NOx as high as 1.25 mg/m$^3$, with a 25% expected increase for doubling the number of vehicles and approximately 50% reduction when considering Euro 6-rated vehicles. The paper demonstrates how the numerical tool can be used to identify the impact of measures to reduce personal exposure, such as protective urban furniture, as traffic patterns and environmental conditions change.

**Keywords:** air quality; nitrogen oxides; dispersion modelling; Computational Fluid Dynamics

## 1. Introduction

The international air quality community, including the World Health Organization (WHO), now recognise that the negative impact of air pollution on people's health is proportional to both pollutants' concentration and time of exposure [1]. Although the health impact remains difficult to establish and more quantitative data are necessary to establish the entity of the problem, this recognition suggests that short-term exposure to high pollutant concentrations—typical at the roadside in urban areas—is dangerous. This is the area of research in which the present paper is positioned. Numerous studies have demonstrated the adverse effect of exposure to nitrogen oxides on human health over periods of both short- and long-term exposure. Over short periods there is significant evidence associating respiratory symptoms [2,3] and systematic support for causality [4,5]. Long-term exposure shows a strong correlation with respiratory and cardiovascular mortality with negative health consequences for children's respiratory system and lung function [6–8]. Even under conditions not exceeding current air quality limits, long-term exposure can result in respiratory symptoms among infants. As a consequence, WHO are reconsidering NOx exposure to account for the difference between ambient and personal exposure [1,9].

The widely established health risks arising due to living in proximity to roads have been the subject of numerous studies [10–15]. In addition to high levels of PM2.5, particularly near roads, dispersion models and experimental studies identify other poisonous

gases such as NOx to be responsible for many adverse effects on health [16,17]. Although NOx plays a key role in the formation of $O_3$ and $PM_{2.5}$ [18], it is common knowledge that the main source of high-level NOx is on-road traffic [19]. Transport is reported to be responsible for 52% of the UK's NOx emission, while 31% is due to road transport [20]. Hence, it is of great importance to measure and manage road traffic-based emissions and protect people from dangerous levels of NOx. Emission and air pollution from traffic is combined with every other source and routinely monitored using Air Quality Monitoring Stations (AQMS), which have been considered to be representative of the air pollution in a provided area [21]. Current AQMS technology, however, cannot capture momentary exposure and, therefore, typically underrepresents actual personal exposure [18,22,23]. Provided the toxicity of the traffic-born pollution, the need for a more accurate and comprehensive measure of exposure is becoming a clear imperative. A great deal of research is being carried out about more accurate and more representative measurement of emissions and air pollution, with particular attention devoted to pollutants' dispersion and also to the feasibility of both passive and active measures to reduce the level of exposure of the most vulnerable people's groups [22,24,25]. The accuracy of AQMS has been challenged in the European air quality project report by Martín et al. [26], where the aim was to evaluate how well AQMS measurements represent the spatial distribution of pollutants. It was concluded that the spatial representativeness of AQMS is low and major attention should be directed towards reducing uncertainties around personal exposure. Sanchez et al. [27] validated a steady-state-weighted CFD model to study the dispersion of NOx against AQMS and a group of deployed NOx sensors. The model was shown to be able to capture NOx dispersion at mesoscale, with the finest resolution of 5 m. Such models can be used to address the representativeness of AQMS locations but cannot capture time-dependent phenomena and personal exposure at a local level. In the same context, Woodward et al. [28] propose a time-based 3D CFD model integrated with a VISSIM traffic model to demonstrate the effect of moving vehicles on air pollution concentration at urban road intersections; in spite of a relatively course computational mesh with the lowest cell size of 0.5 m, the model was able to capture acute NOx concentration events at the roadside, which significantly contribute to public exposure [22] studied NOx levels in an urban area of Madrid, Spain, using CFD in tandem with a VISSIM model to provide high-resolution dispersion estimates of pollutants from road traffic. Hourly and daily temporal resolution, together with the spatial resolution of 3 m, showed that there was up to 80% underrepresentation by the AQMS during the daytime, whereas during the night, the representativeness increased to over 80%. This suggests or rather confirms that AQMS cannot capture the impact of traffic on actual roadside pollution and hence on personal exposure. In the same study, areas with highest personal exposure were reported to be the bus stops and the areas in proximity to traffic lights. In a study by Hess, et al. [29], it has been found that time of day, passengers' waiting location, land use near the bus stop shelters, and the presence of cigarette smoking at bus stops play the most important role in determining the exposure level for pedestrians waiting for a bus. An empirical study carried out by Moore et al. [30] found the bus stop shelter orientation to have a crucial role in personal exposure. It also emphasised the lack of comprehensive bus stop shelter design guidance accounting for traffic levels and shelter orientation and location.

Vegetation has also been reported as an effective passive mitigation strategy for personal exposure to air pollution [31–33]. While the presence of green infrastructure can increase air quality for open-field streets, it may not be as helpful for street canyons where pollutants may be trapped, thereby increasing the concentration at the roadside [33]. Moreover, different types of vegetation have been acknowledged to act as barriers to reducing personal exposure to street-born pollution [34–36]. However, using a CFD dispersion model Xing and Brimblecombe [25] showed that the effect of vegetation might have been overstated. It was shown that the contribution of vegetation to the reduction of air pollution might be insignificant for small-scale green spaces, but also that vegetation can potentially disrupt dispersion, ultimately increasing local concentrations.

In most cases, personal exposure has been observed to be a function of traffic patterns in accordance with the time of the day, wind direction and speed, and configuration of the urban furniture [37,38]. Locally, these variables combine to make exposure a highly transient phenomenon, which can reach—as demonstrated by this study—very elevated momentary levels. In order to measure actual exposure, such observations demand appropriate hardware and an effective number and allocation of sensing elements to realistically capture the complex and time-dependent distribution of pollutants on the street side. Numerical modelling can provide a convenient alternative to field monitoring, but how useful models actually depend on the level of sophistication and resolution and on their validation or verification. While perceived by some to be useful for general planning and air quality policy making, steady-state CFD simulations based on inputs of annual or even hourly pollutant concentrations from routine AQMS are not capable of reflecting the impact of momentary and transient emissions on personal exposure [39,40]. On the other hand, while computationally very efficient, 2D models cannot account for vertical dispersion due to buoyancy and aerodynamic forces, hence cannot capture the full nature of dispersion of traffic-born pollution [41].

Very-high definition, time-based/transient 3D CFD modelling can be conveniently used to simulate the details of pollutants' release, environmental dispersion of gaseous (and solid) matter, and how these processes potentially translate into personal exposure in local urban scenarios. As indicated by the study by Woodward, Stettler, Pavlidis, Aristodemou, ApSimon, and Pain [28], these models must incorporate moving vehicles and realistically account for the modifications they cause to the fluid domain. The major limitation of high-definition 3D CFD models would be the computational cost (hardware, machine time) to simulate adequate periods of physical time. Model development and validation would still require reliable experimental pollutant concentrations, as well as environmental parameters as boundary conditions, but this information can be drawn from previous studies or from simplified, controlled-environment testing, minimising the costs and required effort. A very-high resolution, transient, 3D CFD model can be used to facilitate an improved understanding of the phenomenon of high, momentary exposure to traffic-born emissions in local urban contexts; it would be a valuable tool to generate quantitative information in support of medical studies, and to compare the effects of active (e.g., traffic strategies) and passive (e.g., barriers) air pollution mitigation measures, providing a legitimate basis for policy makers, local authorities, and urban planners to propose changes.

On these grounds, we set out to develop a comprehensive CFD modelling methodology that, supported by generally affordable computational resources, responds to the requirements identified above. To the authors' best knowledge, no other approaches have been proposed that consider the release of pollutant species from the tailpipe of moving vehicles and their dispersion over relatively large urban spaces, using advanced meshing algorithms which enable capturing the necessary fluid-dynamic complexities with very high resolution, while retaining an affordable computational cost. This paper outlines the methodology and demonstrates the capabilities of a very-high-resolution 3D modelling tool, with the aim of supporting a better understanding of urban air pollution at the microscale level and the development of effective exposure mitigation measures.

## 2. Methodology

A comprehensive CFD model has been developed to reproduce a generic open-road urban scenario. A number of vehicles move along a straight road releasing realistic amounts of gaseous emissions from the tailpipe, including nitrogen oxides; depending on the local environmental conditions, the pollutant gases disperse within the main air stream and reach a group of pedestrians on the roadside. The dynamics of the gas dispersion and the levels of momentary personal exposure are captured by the model and reported.

The commercial CFD software Siemens Simcenter STAR-CCM+ has been used to solve the transient, three-dimensional Reynolds Averaged Navier–Stokes (RANS) equations with the Realizable $k$–$\varepsilon$ turbulence closure model over a large computational domain. A

segregated flow solver has been adopted to solve flow equations using a second order upwind discretisation scheme, along with the SIMPLE algorithm for pressure–velocity coupling. All the fluids within the domain are treated as multi-component gases, and the segregated species model is used to solve the continuity equations at the species level. Due to the nature of the gas dispersion, buoyancy has been considered via the application of the Boussinesq model and gravity [42].

A multi-block and locally refined mesh have been used to capture gas dispersion, boundary layer effects, and the jets emerging from the vehicles' exhaust tailpipe. The overset mesh technique has been adopted to enable the simulation of moving vehicles, which are taken as part of a separate computational region exchanging data with the surrounding environment region. The data exchange is realised through the overset mesh, which produces overlapping cells between the two regions as donor and acceptor cells (Figure 1). Refinement of the mesh around the moving objects, along with the requirement of maintaining a similar cell size across the overlapping zone, is extremely important to accurately capture the details of the gas dispersion process. Since the vehicles move along the computational domain emulating moving traffic, the mesh should be suitably refined along the whole vehicle's projected path to capture both the transient nature of vehicles aerodynamics and the exhaust gas dispersion, ensuring mesh compatibility across the donor/acceptor overlapping zone. The Adaptive Mesh Refinement (AMR) method is used in this work to ensure such refinement as the vehicles move; the mesh is refined interactively and not permanently, which eliminates unnecessary refinement (far from the active zone) and reduces the cell count significantly. The flow in the region adjacent to the walls is resolved using the 'Two-Layer All *y*+ Wall Treatment' wall function [42]. Mesh design and generation in that region were carried out in a way to maintain the *y*+ level of the first cell layer (the viscous sub-layer) to less than 5.

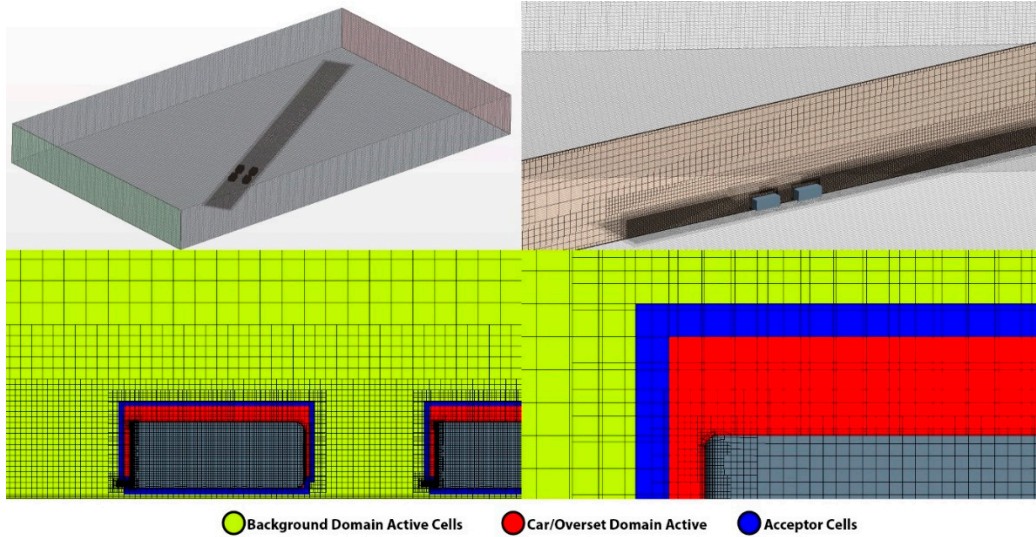

**Figure 1.** Hexahedral multiblock mesh together with Overset mesh configuration around the moving vehicles.

The open-road urban setting modelled in this work is reproduced through a large rectangular box domain of 150 m length, 90 m width, and 15 m height. Details are provided in Figure 2. In this initial phase of the work, which focused on methodology development, the vehicles are represented by boxes with smoothed edges; future work will incorporate more realistic vehicle profiles. A cell base size of 2 m is used for most of the computational domain, while transient refinement focuses on the areas with the largest flow parameters gradients, down to a cell size of 0.01 m (details also provided in Figure 1). The final computational domain included an average of 3.5 M cells in total. In the present study, this newly proposed CFD model has been used to reproduce 10 s of physical time, using

a time step of 0.01 s and 20 internal iterations (per time step). Simulation convergence was determined through the residuals to settle to under $10^{-5}$ for continuity, momentum, and turbulence terms. The simulations have been executed using the High-Performance Computing Cluster available at Oxford Brookes University; by using a 48 computing-core configuration, each simulation ran for approximately 12 h. Siemens Simcenter STAR-CCM+ generally allowed good model scalability; hence, much longer periods of physical time can be simulated using the proposed tool while maintaining affordable computational costs.

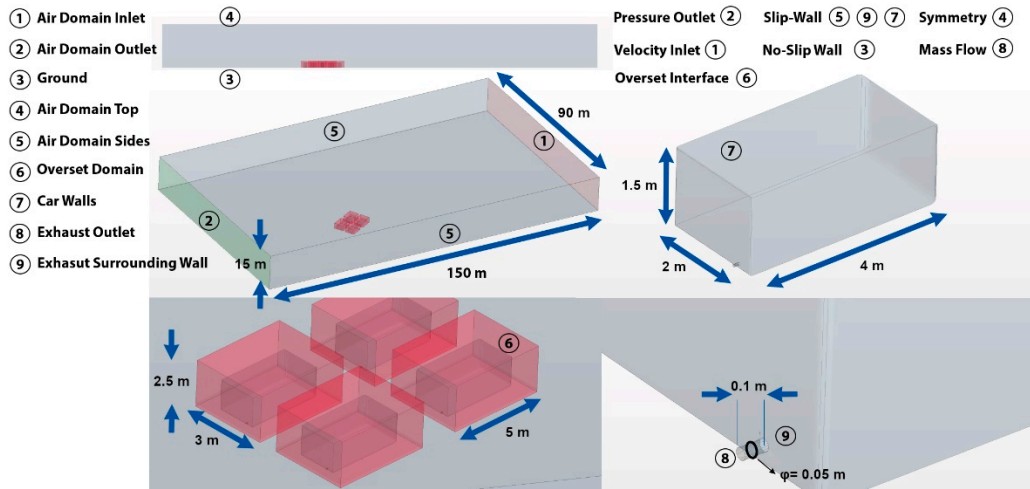

**Figure 2.** Geometry of the computational domain, boundary conditions and relevant dimensions. The top left image shows the wind angle, which is calculated according to a clockwise rotation from the direction of vehicles' travel.

As reported in Figure 2, the computational domain contains a windward surface with 'Velocity Inlet' as the boundary condition, an outlet surface set as 'Pressure Outlet' where the air leaves the defined domain, a ground surface naturally set as a 'No-Slip Wall', and top, left and right surfaces determining the height and width of the domain set as 'Symmetry Plane'. The inlet boundary layers are imposed with the log wind profile commonly used to resolve vertical distribution of wind speed [35]:

$$U = \frac{u_{tau}}{\kappa}\left(\frac{\ln(Z + Z_0)}{Z_0}\right) \tag{1}$$

where Z is the height from the ground, $Z_0$ is the dynamic roughness (0.001 m), $\kappa$ is the Von Karmann constant (0.42), and $u_{tau}$ is friction velocity. The latter is computed based on the reference height $Z_{ref}$ = 10 m and a provided wind speed at the corresponding height of $U_{ref}$ = 5 m/s. Turbulent kinetic energy k and turbulent dissipation rate $\varepsilon$ corresponding to a neutral atmosphere are defined as:

$$k = \frac{u_{tau}^2}{C_\mu^{0.5}} \tag{2}$$

$$\varepsilon = \frac{u_{tau}^3}{\kappa Z} \tag{3}$$

Apart from the exhaust tailpipes, other surfaces of the moving vehicles are set as 'Wall'. The boundary condition for the exhaust tailpipes is 'Velocity Inlet', featuring the relevant gas composition and emerging velocity.

The details of the simulated case studies are summarised in Table 1. The main fluid matter within the computational domain is atmospheric two-component ambient air, with mass fractions of oxygen, nitrogen, and carbon dioxide of 0.233, 0.767 and 0.005, respectively, together with nitrogen dioxide mass fraction of $2.18 \times 10^{-8}$ which is equivalent to a typical background level of $NO_2$ of 0.025 mg/m$^3$, taken from recent Oxford City AQMS

measurements [43]. Ambient temperature and relative humidity are kept constant across all case studies, at 20 °C and 50%, respectively. Ambient air is used as part of the definition of initial and inlet boundary conditions. In all case studies, the wind speed (inlet air velocity) was kept constant at 5 m/s, representing the prevailing UK condition of gentle breeze. The vehicle speed was also kept constant at 20 mph (32 km/h), which is a realistic speed for fluid urban traffic. The vehicles emit hot exhaust gases from the tailpipe at a temperature of 200 °C and at a rate of 28.24 g/s; these levels correspond approximately to vehicles equipped with diesel engines of 2 L capacity, running at an engine speed of 1500 rev/min and assuming a volumetric efficiency of 90%.

**Table 1.** Details of the different case studies.

| Case No. | Number of Vehicles | NOx Emission Rate per Vehicle (as $NO_2$ Equivalent) | Wind Angle (deg) | Mitigation Measure |
|---|---|---|---|---|
| 1 | 4 | 0.04 g/s Euro 5, [44] | 30/60/90 | N/A |
| 2 | 4 | 0.02 g/s Euro 6, [45] | | |
| 3 | 8 | | 30 | |
| 4 | 4 | 0.04 g/s Euro 5, [44] | | Plexiglass barrier by the traffic light |
| 5 | 4 | | | Bus stop shelter facing towards/away the road |

Case 1 is the baseline case study used to demonstrate roadside personal exposure as a result of road traffic. The simulation includes four diesel vehicles releasing an exhaust gas stream into the computational domain through the tailpipe. The multi-component mixture is provided the typical, average diesel exhaust gas composition reported in Table 2 [46]. The vehicles are assumed to be EURO 5 rated, with NOx emissions of 0.04 g/s from the Real Driving Emissions (RDE) work of Costagliola, Costabile, and Prati [44]. This rate is taken as $NO_2$ equivalent for the purpose of modelling. In case study 1, three different wind directions/angles are also modelled to demonstrate the impact on pedestrians' exposure.

**Table 2.** Euro 5 diesel gas components and mass fraction.

| Gas Component | Mass Fraction % |
|---|---|
| $NO_2$ | 0.142 |
| $O_2$ | 15 |
| $H_2O$ | 2.6 |
| $CO_2$ | 7.2 |
| $N_2$ | 75.058 |

Case 2 has the same configuration as Case 1, but the vehicles are assumed to the Euro 6b-rated, releasing NOx at a rate of 0.02 g/s based on the RDE study of Söderena, Laurikko, Weber, Tilli, Kuikka, Kousa, Väkevä, Venho, Haaparanta and Nuottimäki [45]. The tailpipe gas stream composition for $O_2$, $H_2O$, and $CO_2$ remains the same as in Table 2; the mass fraction of $NO_2$ and $N_2$ are recalculated as 0.071% and 75.129%, respectively. Case 3 includes eight EURO 5-rated vehicles and addresses the phenomenon of local, momentary accumulation of pollutant gases before full dispersion, emulating the effects of more realistic patterns of live traffic. Cases 4 and 5 are based again on four EURO 5-rated vehicles and used to demonstrate how barriers can effectively reduce pedestrians' exposure. Case 4 incorporates a plexiglass barrier next to the traffic light where people are closest to the road. Case 5 simulates two common orientations of bus stop shelters, facing towards the road and facing away from it.

## 3. Results and Discussion

This section reviews and discusses the main results of the five case studies outlined in Table 1, with a specific focus on quantifying how NOx emissions from traffic translate into exposure for the public occupying the roadside. For case studies one to four, three individuals stand by a pedestrian-crossing traffic light, including one adult, one teenager, and one child (Figure 3). Monitoring probes are set within the model to measure the NOx concentration at an assumed face height of 1.7 m for the adult, 1.5 m for the teenager, and 1.3 m for the child. The time-resolved levels from these probes are taken as an indication of actual momentary exposure. In case study five, individuals and monitoring probes are located within and immediately outside a bus stop shelter, positioned 15 m before the pedestrian-crossing lights. In case study five, all the probes are located at the same height, representative of an adult's face height (1.7 m).

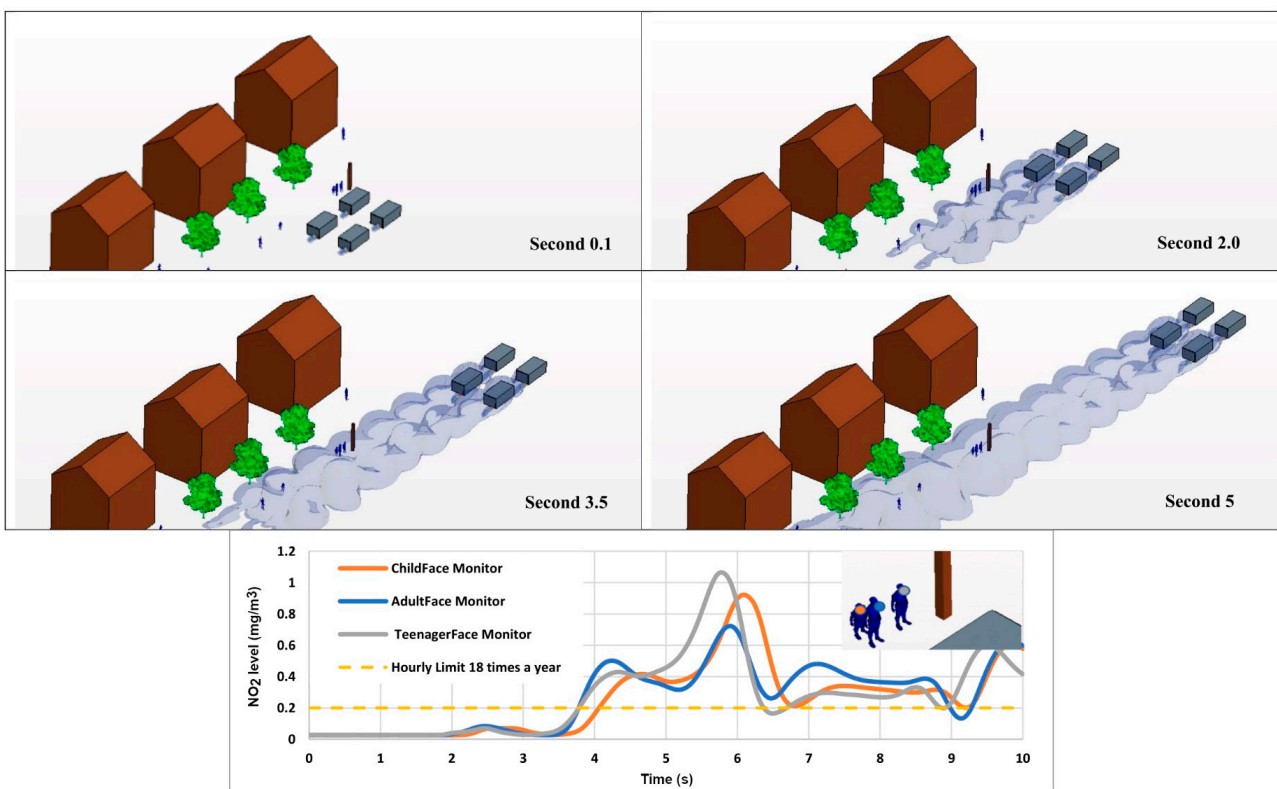

**Figure 3.** $NO_2$ levels for the pedestrians standing by the traffic light.

### 3.1. Case 1. Baseline Model: Demonstration of Personal Exposure and Effects of Wind Direction

Four Euro 5 diesel vehicles move at a constant velocity of 20 mph along a straight road, each releasing NOx (taken as $NO_2$ equivalent for modelling) at a rate of 0.04 g/s. Three individuals of different stature are on the roadside at a traffic light as the vehicles drive past in front of them. Figure 3, which refers to the case of 30 deg wind angle/direction, shows 3D CFD images of $NO_2$ dispersion/distribution within the domain at four relevant times and a plot of $NO_2$ concentrations from the three monitoring probes/individuals as a function of time. The results in Figure 3 indicate that $NO_2$ disperses within the main air stream as the vehicles move forward, reducing its concentration as the wind pushes the gas towards the occupied side of the road and the buildings. The vehicles start moving at 0 s (reaching cruising velocity instantly), and the $NO_2$ concentration at people's breathing height starts rising at 2 s from the background level, showing a multi-modal bell-shaped distribution with the highest peak above 1 mg/m³ at around 6 s, to then decrease once again towards the background level. According to the WHO, average levels of ambient $NO_2$ concentrations should not exceed 0.2 mg/m³ on an hourly basis [47], with an allowance

of 18 times/year exceedance [48]. The results suggest that, while waiting at traffic lights or walking on the footpath of busy roads, pedestrians may experience short periods of exposure to very intense $NO_2$ levels, well over the WHO limits. While momentary exposure should not be compared to hourly-averaged $NO_2$ measurements and associated limits, two considerations appear appropriate at this stage: 1. as reported in the Introduction, there is now a general recognition that short-term high-intensity exposure may pose serious health risks and so high-resolution, high-frequency modelling and measurements are ever more necessary; 2. It is very plausible to assume that, in high traffic conditions (e.g., on a busy road at peak hours), the transient phenomena of pollutants' dispersion to the roadside and momentary exposure may actually repeat continuously for protracted periods of time, and exposure levels may also increase because of localised pollutant accumulation from a greater number of emitting vehicles. This scenario leads to an even greater public health concern. The results presented in this paper and the above considerations also confirm the unrepresentativeness of the AQMS currently distributed in cities, as reported by earlier research [22,49].

From the plot in Figure 3, average $NO_2$ concentration levels can be calculated over the 10 s simulated period. The adult, the child, and the teenager are respectively exposed to average $NO_2$ levels of 0.27, 0.26, and 0.28 mg/m$^3$. These values compare favourably with the $NO_2$ hot spots (roundabouts and crossroads) identified by Santiago, Borge, Sanchez, Quaassdorff, de la Paz, Martilli, Rivas, and Martín [22], where the concentrations vary between 0.08 to 0.4 mg/m$^3$ depending on the time of the day. While an additional process of model validation/verification using field data should be one of the objectives of future work, this favourable comparison provides confidence the model is able to reproduce gas dispersion dynamics realistically. Since the teenager's monitor probe shows the highest concentration of $NO_2$, the same reference point will be taken forward as the basis for comparison in the following case studies.

The results concerning the impact of variable wind angle on actual exposure are reported in Figure 4. The investigation is important because it allows to identify worst-case-scenario environmental conditions for specific urban contexts and assumed traffic patterns. By comparison with the prevailing recorded local conditions, this identification provides, in turn, the ability to assess the actual exposure risk and design more effective mitigation measures considering land use, street configuration, etc. [37,38]. Figure 4 reports 3D CFD images of $NO_2$ gas dispersion for two wind angles, 60 and 90 deg measured from the traffic direction, and $NO_2$ concentrations as a function of elapsed time for all three wind angles. Wind and exhaust gas velocities emanating from the vehicles' tailpipes compose to determine the pattern of $NO_2$ dispersion and the amount that effectively reaches the pedestrians on the footpath. As evident from the 3D images, the net effect of increasing the wind angle is that high pollutant concentrations reach the roadside more quickly, with the dispersed cloud engaging a shorter length of road. The plot shows that increasing the wind angle has the effect of increasing the number of high-concentration peaks, advancing the time and intensity of the first peak. As expected, a more direct wind causes earlier and more in$NO_2$tense momentary exposure. However, during the 10 s simulation, the 90 deg wind also dissipates the pollutants more quickly, and the level of $NO_2$ rapidly settles back to the background level. The average level of exposure reduces from 0.27 mg/m$^3$ for the three pedestrians under 30 deg wind to 0.2 mg/m$^3$ under 90 deg wind.

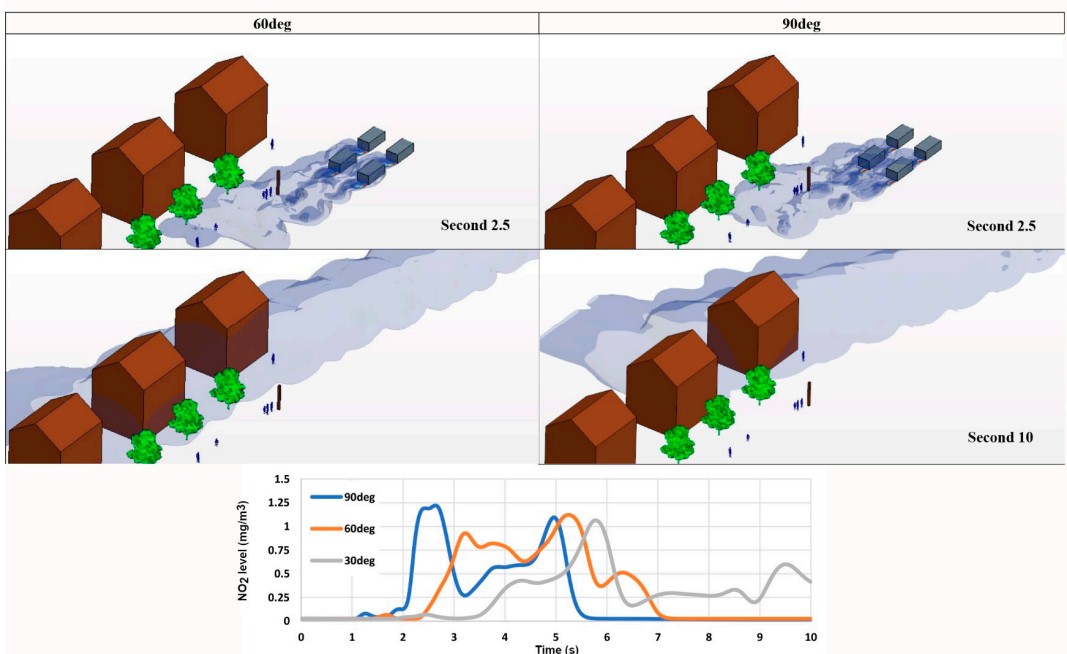

**Figure 4.** NO$_2$ concentrations for the teenager pedestrian standing by the traffic light, from 60 deg and 90 deg wind direction.

### 3.2. Case 2. Personal Exposure from Euro 6-Rated Vehicles

Four Euro 6b diesel vehicles move at a constant velocity of 20 mph along a straight road, each releasing NOx at a rate of 0.02 g/s. In Figure 5, a comparison is drawn between the NO$_2$ concentrations measured by the model probe for the teenager pedestrian (at the height of 1.5 m from the ground) in case studies one and two. The time-resolved concentrations show, as expected, a very similar pattern, with the lower profile reaching as much as 0.2 mg/m$^3$ four seconds after the cars start emitting. The highest peaks appear to be synchronised at second 5.8. Switching, however, from Euro 5 to Euro 6b vehicles produces a reduction in the maximum concentration by approximately 50% to 0.55 mg/m$^3$. The reduction is then sizeable, but the level of personal exposure is still substantial, especially if one considers the additional potential increase due to local pollutant accumulation from a greater number of emitting vehicles. According to Davison, Rose, Farren, Wagner, Murrells, and Carslaw [18], 37% of vehicles on the road in the UK are Euro 5 compliant and 27% Euro 6 compliant, the remainder being categorised as Euro 4-2. Considering that most of the UK fleet is Euro 5 or below, roadside emissions pose a significant risk to the health of pedestrians.

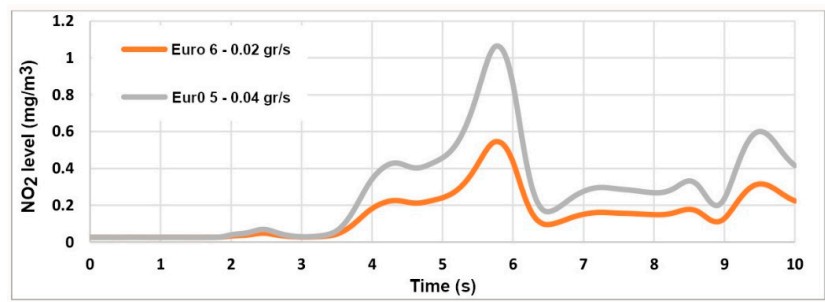

**Figure 5.** Comparison of NO$_2$ concentrations for the teenager pedestrian standing by the traffic light, for two different levels of tailpipe emissions.

### 3.3. Case 3. Increase in Exposure Due to Localised Pollutants Accumulation (8-Vehicle Model)

The effect of localised accumulation of pollutants is demonstrated in Figure 6, where a set of four Euro 5 diesel vehicles is followed by another analogous set at a distance of 18 m, equivalent to a two-second time distance between on-road vehicles. The plot in Figure 6 compares the time-resolved $NO_2$ concentrations measured by the model probe for the teenager in case studies one and three. The 8-vehicle model profile shows two major peaks of exposure. The first peak is almost perfectly aligned with the one from the 4-vehicle model, just slightly lower, and corresponds to the emissions from the first set of vehicles. The 3D CFD images reported in Figure 6 show how the second set of vehicles shields the pedestrians from the emissions of the first set and hence partially disperses the gases, leading to marginally lower values of $NO_2$ in the first peak. However, the second set of vehicles add more to the existing $NO_2$ concentrations, generating a localised, albeit temporary, accumulation and producing a second peak at a higher level in the wake of their movement. The average value of $NO_2$ concentration to which the teenager is exposed rises from 0.28 mg/m$^3$ for the 4-vehicle model to about 0.42 mg/m$^3$ for the 8-vehicle model. An increase in exposure is also experienced by the other pedestrians standing by the traffic light, with the adult pedestrian subject to the greatest increase (Figure 7).

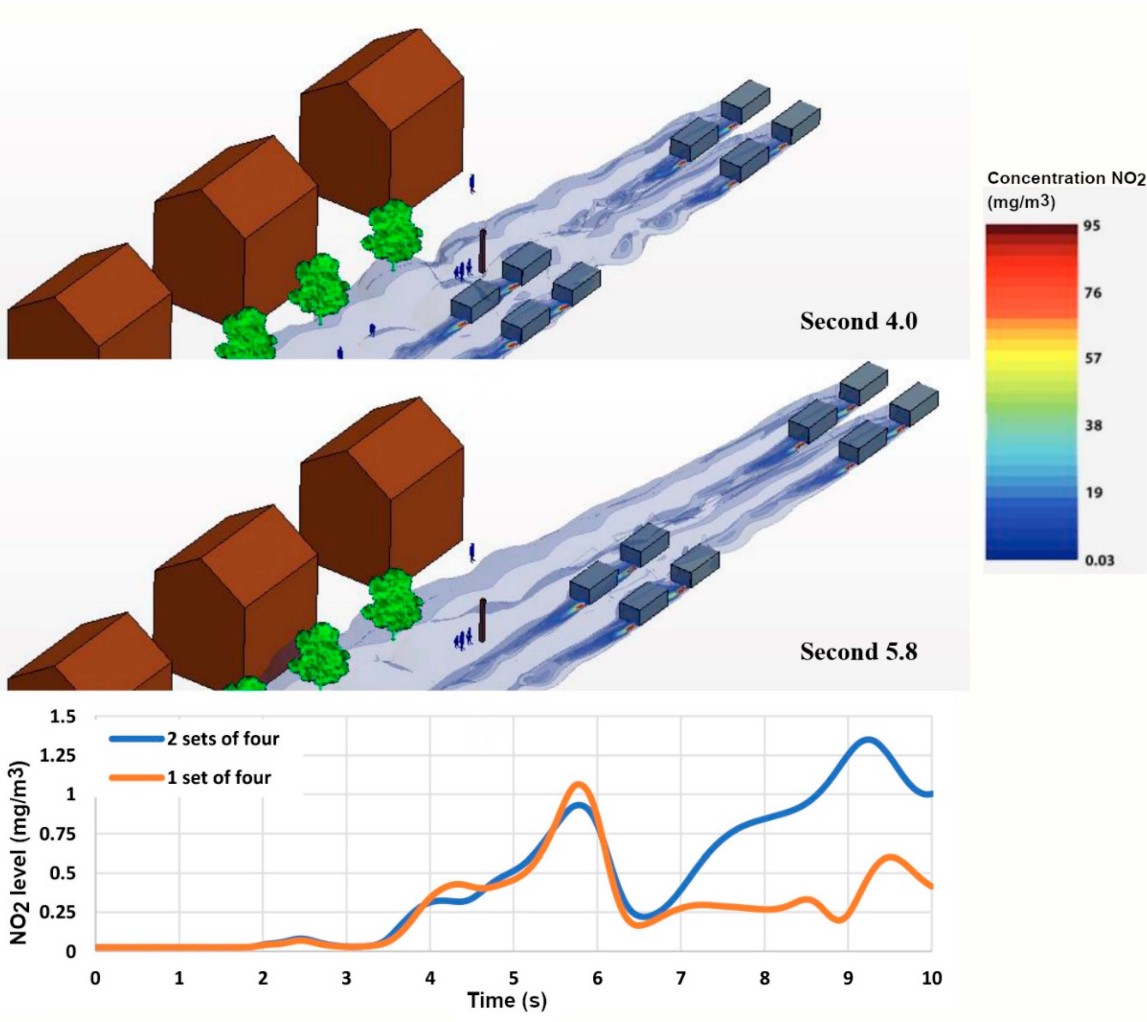

**Figure 6.** Comparison of $NO_2$ concentrations from one and two sets of Euro 5 vehicles, for the teenager pedestrian.

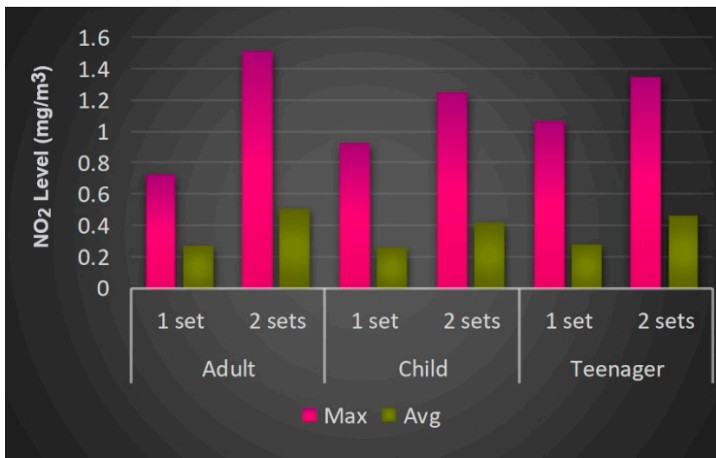

**Figure 7.** Maximum and average levels of exposure to $NO_2$ for the three pedestrians at the traffic light—one set of vehicles vs two sets.

### 3.4. Case 4. Effect of Potential Intervention Measures

In order to reduce personal exposure to air pollution at critical spots such as traffic lights, bus stop shelters, or transport hubs, mitigation measures can be put in place. The simplest measures are passive, consisting, for example, of barriers that may shield pedestrians from traffic-born pollutants in the proximity of traffic lights. The high-resolution CFD tool presented in this paper can be used effectively to support the design and virtual evaluation of mitigation measures for specific urban settings under prevailing local environmental conditions. In this case study, four Euro 5 diesel vehicles move at constant velocity along a straight road, each releasing NOx at a rate of 0.04 g/s. A very simple shield of plexiglass has been designed and incorporated into the model to examine the potential effect of such intervention, Figure 8.

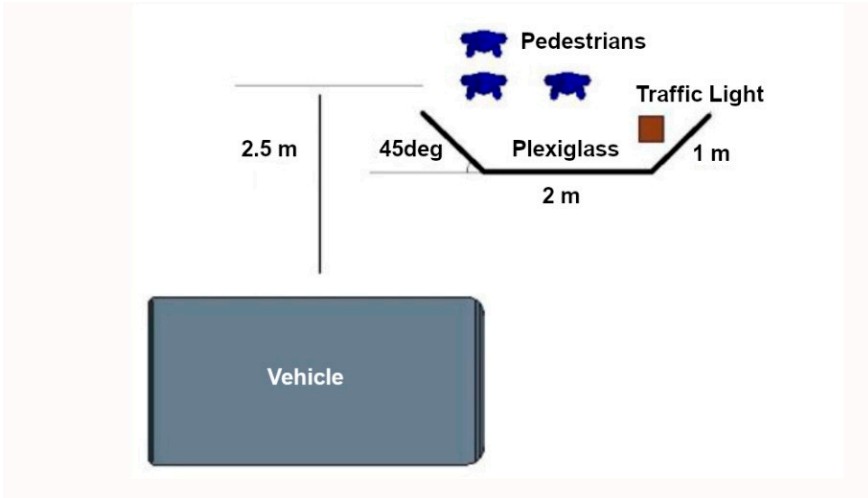

**Figure 8.** Schematic design of the plexiglass shield used by the traffic light (Top View).

The main results, presented in Figure 9, indicate that the barrier significantly reduces the exposure levels, removing the peak of $NO_2$ concentration within the 10 s simulated period. By using this approach, the average exposure for the teenager pedestrian is reduced by 50%, down from 0.28 mg/m$^3$ to 0.14 mg/m$^3$. Similar exposure reductions are experienced by the other two pedestrians.

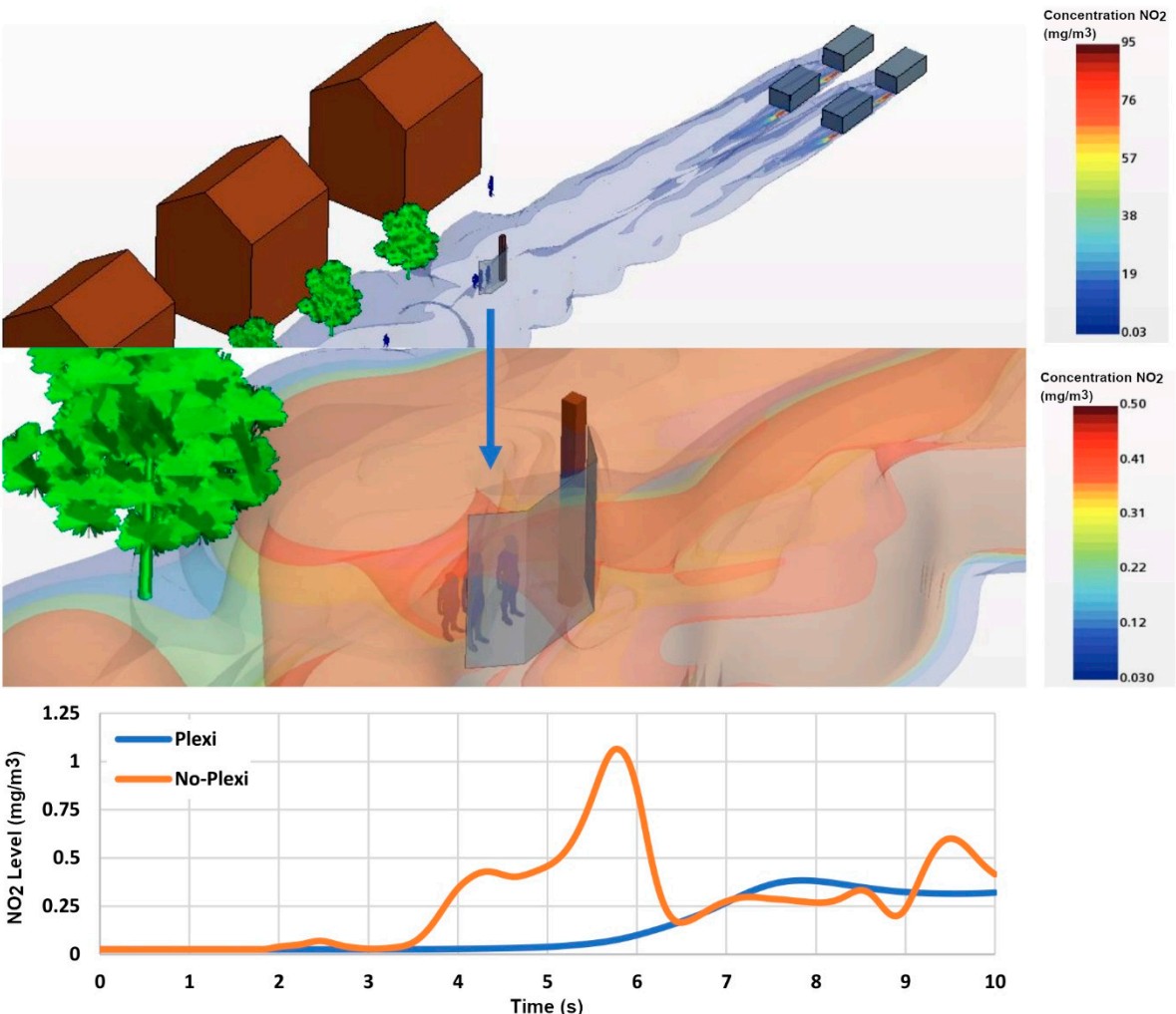

**Figure 9.** Exposure reduction due to the addition of a plexiglass barrier at the traffic light.

### 3.5. Case 5. Bus Shelters

In this case study, the same single set of Euro 5 category vehicles has been used to compare personal exposure for people standing inside and outside a bus shelter of generic configuration. The orientation of the bus shelter, i.e., facing towards (FT) or facing away (FA) from the road, has also been investigated. The latter is reported to be a crucial factor determining the personal exposure of occupants [29,30].

Figure 10 shows 3D CFD images of $NO_2$ distribution in and around the bus shelter for both the FA and FT orientations. Figure 11 reports the time-resolved $NO_2$ concentrations to which three adult individuals are exposed, with persons P1 and P2 standing inside the shelter and person P3 standing outside. The results of the simulations clearly demonstrate that bus shelters offer some level of protection, with P3 consistently exposed to higher levels of $NO_2$ compared to those standing inside (P1 and P2), regardless of bus shelter orientation. However, the simple shelter design considered here does not feature lower or upper apertures along the side and back panels, typically seen in real shelter designs. This leads to some time-dependent pollutant accumulation, as evidenced by the rising profiles of the P1 and P2 monitors in FA orientation. As clear comparing the results for the FA and FT orientations in Figure 11c, the shelter orientation plays a significant role in terms of potential exposure reduction. On average, those standing inside the FT bus shelter suffer 3.8 times more $NO_2$ exposure than those standing in the FA one (0.03 mg/m$^3$ vs. 0.114).

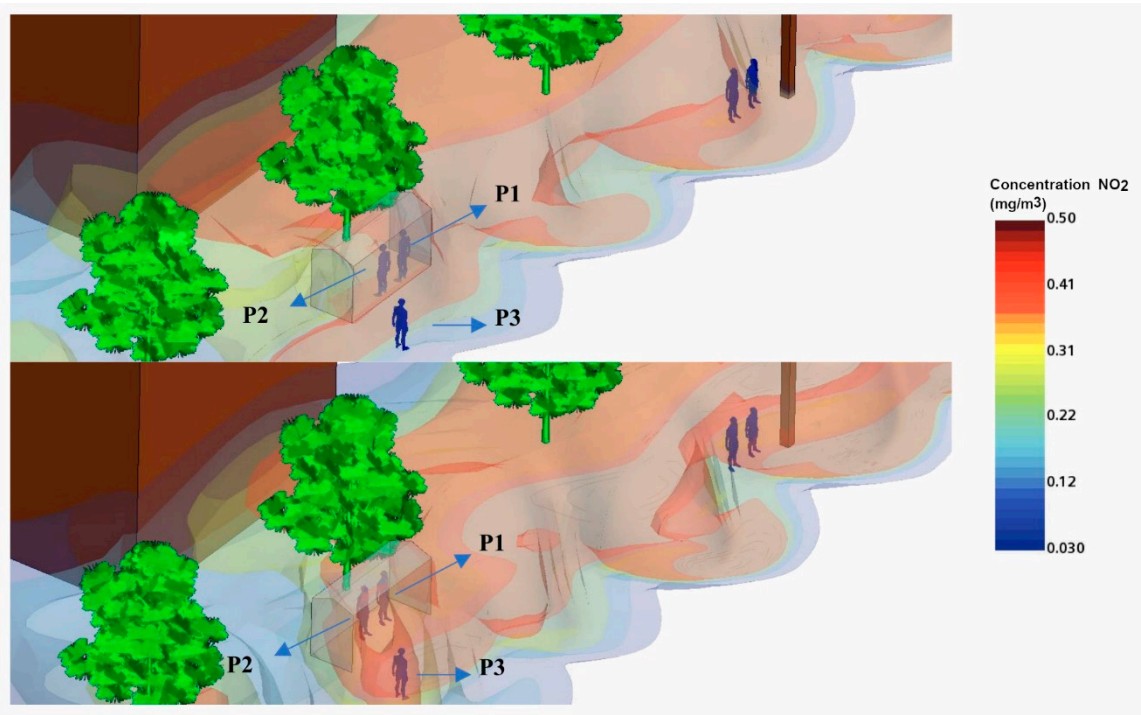

**Figure 10.** Illustration of NO$_2$ gas dispersion for the FA and FT orientation of a generic bus stop shelter, located 15 m downstream of the traffic light.

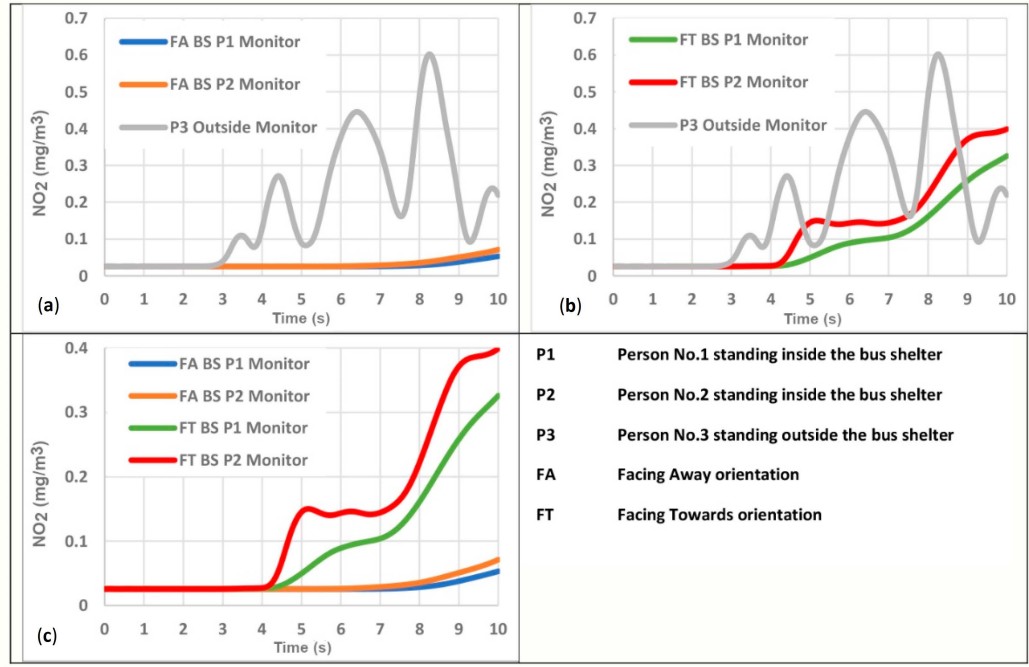

**Figure 11.** NO$_2$ levels for (**a**) people inside and outside facing away bus shelter (FA BS), (**b**) people inside and outside of facing towards bus shelter (FT BA), and (**c**) people inside FA BS and FT BS.

The average concentration level for P3 is measured to be 0.287 mg/m$^3$. The results presented here are limited to the case of a semi-open street, which allows traffic-born pollution to continue to disperse past the immediate roadside and potential bus stop locations. The impact of bus stop shelters on exposure changes in different street configurations is also affected by the land use, as reported by Hess, Ray, Stinson, and Park [29].

## 4. Conclusions

This paper employs a 3D Computational Fluid Dynamics model to perform high-resolution temporal and spatial simulations of pollutant gas dispersion in local urban contexts. While retaining computational efficiency thanks to the deployment of the Overset Meshing and Adaptive Mesh Refinement techniques to minimise the mesh count, the 10-cm-mesh-resolution approach enables a realistic account of how the pollutant gas emitted from the tailpipe of moving vehicles translates into momentary, high-intensity exposure for the public on the roadside. A single set of four Euro 5-rated diesel vehicles travelling at a constant speed of 20 mph in a semi-open straight road may lead, depending on wind conditions, to exposure to $NO_2$ peaks of 1.25 mg/m$^3$, with 10 s average exposure levels between 0.2 and 0.28 mg/m$^3$. Two sets of similar vehicles, used to represent a more realistic traffic pattern, generate two peaks of roadside exposure, with the second peak approximately 25% higher due to temporary gas pollution accumulation at a local level. These results suggest that in densely trafficked urban roads—e.g., main city arteries at peak times—momentary high-intensity exposure events may, in fact, recur continuously for protracted periods of time, outlining a very serious and previously underestimated public health concern. Depending on their location, frequency of measurement, and data reporting setup, routine roadside monitoring stations cannot fully capture the transient gas dispersion phenomena associated with moving traffic, indeed underestimating the problem of personal exposure, as indicated in previous research.

While the average fleet vehicle across the globe becomes increasingly cleaner, through a process driven by restrictive emission regulations, increasing taxation, and forthcoming bans on traditional internal combustion engines, actual vehicles on the road today are still high emitters, with the vast majority of the UK fleet rated Euro 5 or below. Exacerbated by non-exhaust emissions, air pollution from traffic will continue to pose a significant health risk for the foreseeable future. An increased effort must therefore be made to better understand the dynamics of personal exposure and develop feasible public protection strategies. The 3D CFD modelling approach proposed in this paper is flexible in terms of urban geometry, traffic pattern, and environmental boundary conditions and provides an effective virtual testing platform to assess the impact of both active and passive measures to minimise exposure. The exemplary results from the investigation show that installing a simple plexiglass barrier at pedestrian crossings may eliminate the momentary peaks of exposure to traffic-born NOx, reducing the average level by 50%; they also show that changing the orientation of bus stop shelters—from facing the road to facing away from it—significantly increases the level of protection they offer.

**Author Contributions:** Methodology, S.T., F.B.; validation, S.T.; formal analysis, B.M.; resources, D.M.; writing—original draft preparation, S.T., F.B.; writing—review and editing, B.M., D.M.; project administration, F.B. All authors have read and agreed to the published version of the manuscript.

**Funding:** This research received no external funding.

**Institutional Review Board Statement:** Not applicable.

**Informed Consent Statement:** Not applicable.

**Data Availability Statement:** Not applicable.

**Conflicts of Interest:** The authors declare no conflict of interest.

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
