# Peer review of "Simulation of Traffic-Born Pollutant Dispersion and Personal Exposure Using High-Resolution Computational Fluid Dynamics"

_environments, doi:10.3390/environments9060067_

Round 1
Reviewer 1 Report
In this study, the traffic-born pollutant dispersion and personal exposure using high-resolution computational fluid dynamics were simulated and investigated by Tajdaran et al. This study seems very interesting. However, some modifications are necessary.:
- There are a lot of error “Error! Reference source not found” in the article, which need to be revised.
- The description of the simulation experiment on CFD is not specific enough, and detailed simulation experiment details are needed, such as the mesh number.
- How is the data in Table 2 obtained?
- The order of Figure 6 is incorrect.
- The formation mechanism of NOx needs to be further discussed. These literatures are helpful to analyze the formation mechanism of NOx emissions:Comparative study of pilot–main injection timings and diesel/ethanol binary blends on combustion, emission and microstructure of particles emitted from diesel engines. Optimization of palm oil biodiesel blends and engine operating parameters to improve performance and PM morphology in a common rail direct injection diesel engine
Author Response
1. Yes done. Unfortunately, the error was caused by word to pdf conversion, and it is now resolved.
2. The simulation methodology together with meshing approach is given in details under the 'Methodology' section. Mesh base cell size together with total cell count is specifically outlined in lines 188-190.
3. The table data is taken from the book 'basic of combustion engines' cited as reference number 46 in line 236-237.
4. Yes thanks, it's been corrected. Again, the conversion from word to pdf issue and it is resolved now.
5. The scope of this study is to model the dispersion of NOx emission rather than discussing the roots and reasons of why NOx is produced. The paper is concerned with the dispersion modelling at high-resolution to understand the dynamics of personal exposure and develop feasible public protection strategies.
Reviewer 2 Report
Due to the under-representativeness of road side monitoring stations, the authors present a comprehensive modeling study of personal exposure to road traffic emissions of various rates and intensities from moving vehicles under different wind fields as well as examine the role of mitigation measure in protecting pedestrians from momentary high pollutant levels. The results are straightforward and interesting. There are a few details that need to be clarified before publication.
Major:
- There are a lot of “Error! Reference source not found.” appearing in the text. Please fix them.
- Do the authors show schematically where the bus stop shelter is in Figure 2? I cannot imagine where the bus stop shelter is when the authors said “positioned few meters before the pedestrian-crossing lights” in Lines 274-275. Please direct audience to your schematic figures appearing later in the manuscript at an early point or better describe it.
- I am curious why in Case 5 the authors put the probes at the same height representative of adult’s face height instead of at different heights for different individuals as the other cases. I am just curious if the mitigation measure of setting up a bus stop shelter protect different individuals to a similar degree.
- I wonder if the authors define the duration of the vehicle stop at the traffic light. And if the emission rates change when the vehicles stop?
- I wonder if the 3 individuals stand upwind or downwind of the emissions. Please clarify. And if the direction of the individuals’ faces matter? Does it always mean that when the pedestrians face the roadside, they are exposed to higher levels of pollutant? I guess the authors do not consider the air dynamics associated with human.
- In Lines 332-333: Do the authors have a possible explanation why increasing the wind angle would have the effect of increasing the number of high-concentration peaks, as shown in Figure 4?
- In Figure 4, do the authors quantify the total amount of pollutants that people can inhale? For wind angle of 90, people seem to be exposed to higher concentration of pollutant but for a shorter duration; for wind angle of 30, people seem to be exposed to lower concentration of pollutant but for a longer duration. The dose of pollutant people inhale depends on the concentration of pollutant as well as the exposure time.
- It looks like that the authors do not consider the air dynamics associated with buildings off the road. It is an open urban scenario instead of a street canyon scenario.
Minor:
- In Line 274, an “a” is missing before word “few”.
- Please add a time stamp in the lower left panel of figure 4.
Author Response
- Fixed thanks. Unfortunately, the error was caused by word to pdf conversion, and it is now resolved.
- The bus shelter is positioned 15m downstream of the traffic light. It is now reflected both in text (line 268) and in caption of figure 10.
- The scope of the overall study is to address the practicality of our high-resolution CFD model of dispersion. For this reason, several different approaches are trialled. Whilst we use probes at different heights for cases 1-4, in case 5 the aim was to look at the effect of bus stop shelter orientation and the protection that such shelters can provide against road emission. We will go to further details in the next paper.
- Whilst the authors appreciate the differences that driving conditions make on the emitting levels of pollution, the focus of the work was not to explore the various level of emission across different driving conditions but rather to consider a realistic level of NOx emission from a moving constant speed vehicle taken from the literature experimental tests and simulate the NOx dispersion in high-resolution in order to capture momentary exposure peaks which pedestrians experience on the roadside.
- The direction of the wind is specified in Figure 2. The individuals stand downwind being therefore exposed to emissions. Exposure depends on the wind angle as is also addressed in the literature review section. Details of the parametric studies and variables are outlined in Table 1.
- Lines 332-333 are preceded by the explanation of why the pedestrians experience higher exposure at 60 and 90deg (lines 325-329).
- That is exactly true which is why it has been explained in lines 331-332 that the average level during the 10 second exposure is smaller under 90deg wind direction. This provides the insight for further health research studies to look at both scenarios i.e. very short time high exposure versus longer intervals with lower exposure levels.
- This paper is concerned with open road urban scenario chosen to reveal that even well-ventilated roads cannot protect pedestrians against high level exposure to traffic born pollutions. However the effect of buildings and vegetation on pollution concentration is addressed in the background/literature review section.
Minors:
- Resolved
- added
Round 2
Reviewer 1 Report
The authors made a lot of changes according to my suggestions. Therefore, the current format can be accepted.
Reviewer 2 Report
The authors well addressed all comments and questions.